

# Comparison of animal models for immune premature ovarian insufficiency

Anchun Hu[1,2,*], Yanli Mu[2,*], Guanyou Huang[2] and Shuyun Zhao[2]

[1] Reproductive Medicine Center, Xingyi People's Hospital, Qianxina Prefecture, Guizhou, China
[2] Department of Obstetrics and Gynecology, School of Clinical Medicine, Guizhou Medical University, Guiyang, Guizhou, China
[*] These authors contributed equally to this work.

## ABSTRACT

Premature ovarian insufficiency (POI) severely impacts women's reproductive and overall health, yet effective treatments remain elusive. Research on its pathogenic mechanisms and therapeutic strategies is therefore critical. Due to the scarcity of ovarian samples from POI patients, animal models have become indispensable tools for investigation. Notably, immune-related POI accounts for an increasing proportion of cases, with over half of idiopathic POI cases hypothesized to involve immune dysregulation. Consequently, immune-mediated POI animal models are widely used to study immune-related mechanisms. This article compares the advantages, limitations, and applications of various immune-related POI animal models, aiming to guide researchers in selecting the most appropriate model for their specific research goals and experimental designs.

## INTRODUCTION

Currently, global fertility rates are rapidly declining, posing unprecedented challenges and crises (*GBD 2021 Fertility Forecasting Collaborators, 2024*). In females, the total number of ovarian follicles gradually decreases with age after birth, and reproductive capacity—encompassing both endocrine and reproductive functions—undergoes irreversible decline after sexual maturity (*Harasimov et al., 2024*). Premature ovarian insufficiency (POI), previously termed premature menopause, primary ovarian insufficiency, or premature ovarian failure (*Nash & Davies, 2024*; *Touraine et al., 2024*), is defined as ovarian dysfunction characterized by menstrual irregularities and elevated follicle-stimulating hormone (FSH) levels in women under 40 years of age. In 2016, the European Society of Human Reproduction and Embryology (ESHRE) formally standardized the term "premature ovarian insufficiency" (POI) to replace previous nomenclature (*Webber et al., 2016*). However, despite this standardization, significant controversies and heterogeneity persist regarding the specific diagnostic criteria and classification of POI, such as divergent FSH threshold, ambiguous role of Anti-Müllerian hormone and inconsistent definitions of menstrual dysfunction. The prevalence of POI ranges from 0.5% to 4% (*Touraine et al., 2024*).

Corresponding authors
Anchun Hu, 531507839@qq.com
Shuyun Zhao, zhaoshuyun-sci@126.com

As a critical organ for both reproduction and endocrine regulation, ovarian dysfunction in POI not only severely impacts fertility—leading to infertility, menstrual disorders, and reduced sexual function—but also increases long-term health risks, including cardiovascular disease, osteoporosis, and cognitive decline (*Okoth et al., 2020*; *Jones et al., 2020*; *Sochocka et al., 2023*). Etiologies of POI include iatrogenic factors (*e.g.*, ovarian surgery, pelvic/abdominal radiotherapy/chemotherapy), X chromosomal abnormalities, genetic mutations, autoimmune disorders, and other unidentified causes (*Nash & Davies, 2024*; *Touraine et al., 2024*).

Notably, POI patients exhibit a higher susceptibility to autoimmune diseases, with 10%–30% of cases coexisting with autoimmune conditions, particularly thyroid and adrenal disorders (*Ishizuka, 2021*; *Domniz & Meirow, 2019*). Autoantibodies are detectable in 40%–50% of POI patients (*Domniz & Meirow, 2019*; *Belvisi et al., 1993*), suggesting immune dysregulation as a key contributor to at least half of idiopathic POI cases. Although ovarian biopsy remains the theoretical gold standard for autoimmune-POI diagnosis, its invasive nature and potential to damage ovarian tissue make human ovarian tissue or oocyte procurement ethically and practically unfeasible, thus rendering it inappropriate for routine diagnosis. The non-invasive markers (elevated FSH, low anti-Müllerian hormone (AMH), reduced follicle count on ultrasound) are crucial indicators of diminished ovarian reserve and functional decline, forming the cornerstone of clinical diagnosis. However, these markers reflect the functional endpoint of POI rather than its specific underlying etiology. They cannot reliably distinguish autoimmune-POI from other causes of ovarian failure. Consequently, rodent models (mice or rats) are widely utilized to investigate POI pathogenesis, endocrine-metabolic alterations, and therapeutic interventions. In autoimmune POI research, three primary modeling strategies are predominantly employed: (1) active immunization induction through zona pellucida glycoprotein 3 immunization, which simulates antibody-mediated ovarian damage; (2) passive immunization employing adoptive transfer of autoreactive T-cells to model cell-mediated autoimmune responses; and (3) gene-edited models such as autoimmune regulator (AIRE)-deficient mice that develop spontaneous POI. Each approach offers distinct advantages while presenting specific limitations in replicating the full spectrum of human disease manifestations.

While rodent models cannot fully replicate human autoimmune POI complexity, they offer valuable translational insights through conserved immunological pathways. These models are crucial for studying ovarian damage mechanisms and testing initial therapies, but their artificial induction, absence of human menstrual cyclicity, and inability to model polygenic/environmental interactions limit their representativeness of spontaneous human POI. Translational challenges include: (1) physiological disparities in reproductive biology (*e.g.*, folliculogenesis dynamics, menstrual cycle absence); (2) etiological oversimplification (single-mechanism induction *vs.* human polygenic/environmental interactions); (3) therapeutic translation barriers due to interspecies differences; (4) inability to mirror clinical heterogeneity. Despite these constraints, rodent models remain indispensable for mechanistic studies and preclinical screening, enabling controlled experiments impossible in humans. They illuminate disease components rather than the complete human condition. This article aims to compare existing autoimmune POI animal models, providing

researchers with evidence-based guidance for selecting optimal modeling approaches tailored to specific study objectives.

## SURVEY METHODOLOGY

To systematically compare the animal models for immune premature ovarian insufficiency, this PRISMA-guided review searched PubMed, Medline and Embase without date restrictions, yielding literature from 1965–2025. The search strategy included combinations of: ("immune premature ovarian insufficiency" OR "POI/POF [premature ovarian insufficiency/failure]" OR "immunity and ovarian dysfunction" OR "Experimental autoimmune oophoritis" OR "experimental ovarian autoimmunity") AND ("animal models" OR "mice" OR "rat"). For specific model evaluation, additional terms were used ("ZP3", "ZP", "ovarian antigens", "Neonatal thymectomy", or "inhibin-α", "rag", "aire", or "nude mice", "passive transfer of autoantibodies"). Editorials, letters to the editor, and case reports were excluded from this review.

### Animal models

The pathogenesis of autoimmune-related POI involves the breakdown of immune tolerance, leading to the loss of the body's ability to distinguish self-ovarian tissues. This triggers autoimmune inflammation and immune responses. Both humoral immunity (*e.g.*, autoantibody production) and cellular immunity (*e.g.*, T-cell dysfunction) are closely associated with the development of autoimmune POI (*Kirshenbaum & Orvieto, 2013*).

Current methods for constructing immune-mediated POI animal models include the following: (1) active immunization with ovarian-specific antigens: zona pellucida 3 peptide (pZP3), crude ovarian antigens, zona pellucida 4 peptide (pZP4); (2) neonatal thymectomy in animals: surgical removal of the thymus in newborn rodents to disrupt immune tolerance; (3) inhibin-α-induced autoimmune targeting of the pituitary-ovarian axis; (4) gene-edited models: Rag gene knockout (*e.g.*, Rag1$^{-/-}$ or Rag2$^{-/-}$ mice), AIRE gene knockout (mimicking autoimmune polyendocrine syndrome type 1), knockout of other immune-related genes (*e.g.*, FoxP3, BNDF); (5) adoptive transfer nude mouse models: transfer of autoreactive T cells into immunodeficient nude mice to study ovarian-specific immune damage; (6) passive transfer of autoantibodies: injection of autoantibodies (*e.g.*, anti-ZP3 or anti-FSH receptor antibodies) to induce ovarian dysfunction; (7) other potential target antigens: candidate antigens for POI induction, including:

3 beta-hydroxysteroid dehydrogenase (3β-HSD), Heat-shock protein 90-beta (HSP90β); HPV4 (exploring cross-reactivity hypotheses between viral proteins and ovarian antigens).

These approaches provide versatile platforms for studying immune mechanisms, therapeutic interventions, and gene-environment interactions in POI. While these animal models have provided valuable insights, it should be noted that many were developed decades ago and lack systematic validation using modern immunological standards. Contemporary techniques such as CRISPR-based gene editing, single-cell RNA sequencing, and high-dimensional immune profiling could significantly improve model characterization. For instance, traditional approaches like neonatal thymectomy and passive antibody transfer have seen declining use, being largely superseded by more precise genetic

and cellular manipulation methods that better mimic human disease mechanisms while reducing off-target effects. Moving forward, incorporating these advanced methodologies will be crucial for developing more physiologically relevant models with greater translational potential for POI research.

## Active Immunization with Ovarian Autoantigens
### pZP3 immunization

The zona pellucida (ZP), a glycoprotein layer surrounding mammalian oocytes, serves as an ovarian-specific target antigen and plays a critical role in oogenesis. In mice, ZP glycoproteins are synthesized exclusively in oocytes (*Moros-Nicolás et al., 2021*) and consist of three glycosylated proteins: ZP1, ZP2, and ZP3 (*Bleil & Wassarman, 1980*). ZP2 and ZP3 exist as 1:1 monomers, accounting for over 80% of total ZP protein and are essential for ZP assembly. ZP1 forms dimers and constitutes less than 20% of murine ZP protein (*Litscher & Wassarman, 2020*). ZP genes are expressed solely in growing oocytes of female mice (*Wassarman & Litscher, 2018*). ZP1 maintains the structural integrity and matrix of the zona pellucida (*Rankin et al., 1999*), while ZP2 knockout results in a thinner ZP that disappears post-ovulation (*Rankin et al., 2001*). ZP3 knockout completely prevents ZP formation (*Liu et al., 1996*; *Rankin et al., 1996*), and ZP3 mutations can induce oocyte developmental abnormalities, such as zona-free oocytes (*Cao et al., 2020*) or empty follicle syndrome (*Chen et al., 2017*).

ZP3 is central to murine ZP development, requiring at least two glycoproteins (Zp1-Zp3 or Zp2-Zp3 combinations), with Zp3 being indispensable (*Dean, 2004*). A single copy of the Zp3 gene is uniquely transcribed in growing oocytes, and ZP proteins are detectable in ovaries within three days postpartum (*Rhim et al., 1992*). Studies reveal that ZP3 mRNA levels significantly exceed those of other ZP genes across all follicular stages (primordial, primary, secondary, antral, and preovulatory) (*Zhang et al., 2018*), directly linking ZP3 to zona pellucida synthesis and oocyte maturation (*Sun, Liu & Kikuchi, 2008*).

As early as 1992, researchers demonstrated 67% homology between murine and human ZP3 proteins. Immunization with pZP3 induces anti-ZP3 antibodies in mice, which bind ovarian ZP3, triggering autoimmune oophoritis (*Rhim et al., 1992*), oocyte destruction, follicular depletion, and amenorrhea (*Yin et al., 2024*). This homology enables ZP3 to replicate human POI phenotypes, making it a key antigen for modeling autoimmune POI in mice.

Modeling method using ZP3 as follows: the amino acid sequence of the synthetic ZP3 330-342 peptide is NSSSSQFQIHGPR. The synthesized peptide powder is dissolved in ddH$_2$O to prepare a ZP3 peptide solution (one mg/mL), which is then emulsified 1:1 by volume with Freund's complete adjuvant (containing Mycobacterium tuberculosis components) until a stable water-in-oil emulsion forms. The emulsion (0.1–0.15 mL) is administered *via* subcutaneous multi-point injection (hind paws, abdomen, and back) or intraperitoneal injection. Two weeks later, a secondary immunization is performed by emulsifying the ZP3 solution with Freund's incomplete adjuvant (without M. tuberculosis components) and injecting it at the same sites (*Rhim et al., 1992*; *Yin et al., 2024*; *Xie et al., 2024*). Some studies suggest that a third booster immunization after an additional

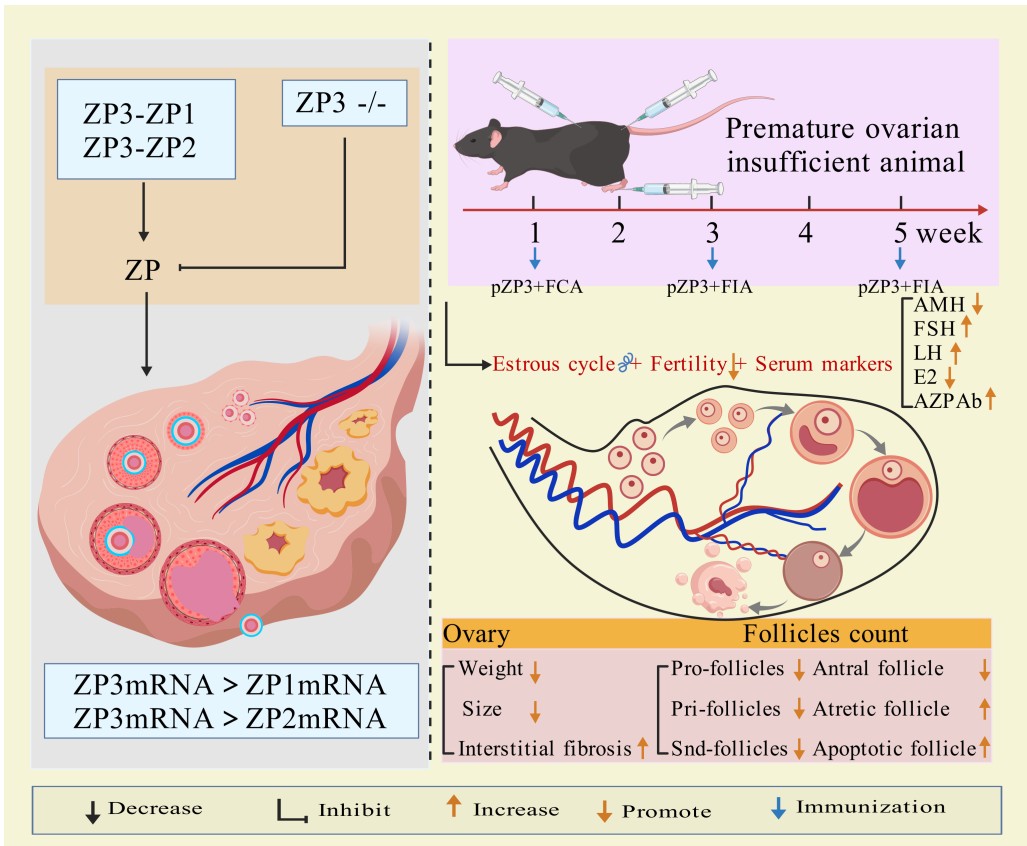

**Figure 1** **The mechanisms and application of pZP3-induced POI.** Created with BioRender.com.

2-week interval improves modeling efficacy (*Wang, Chen & Lu, 2017*). Elevated anti-zona pellucida antibodies, detectable in the animals for weeks after the final injection, along with histopathological evidence of ovarian lymphocyte infiltration, irregular estrous cycles, hormonal imbalances, and reduced follicular development, confirm successful establishment of the premature ovarian insufficiency (POI) model (*Rhim et al., 1992*; *Yin et al., 2024*; *Xie et al., 2024*; *Wang, Chen & Lu, 2017*). The mechanism of ZP3 action is illustrated in Fig. 1.

pZP3-induced autoimmune ovarian disease (AOD) is characterized by lymphocyte infiltration, elevated serum/ovarian anti-ZP antibodies, ovarian atrophy, and non-infectious oophoritis (*Zhang et al., 2019*). This model is widely used to study T-cell-mediated autoimmune mechanisms, including: effects of human amniotic epithelial cells or placental mesenchymal stem cells on splenic regulatory T cells (Tregs) and ovarian function (*Zhang et al., 2019*; *Yin et al., 2018*); post-transplant changes in CD8+CD28− T cells, interleukin-10 levels, Th1/Th2 balance, and uterine natural killer (uNK) cell activity (*Yin et al., 2024*; *Lu et al., 2019*); therapeutic mechanisms of exosomes from bone marrow mesenchymal stem cells in reducing granulosa cell apoptosis/pyroptosis (*Xie et al., 2024*); molecular mechanisms of traditional Chinese medicine in autoimmune POI (*Chen et al.,*

*2021*; *Chen et al., 2022*). Figure 1 illustrates the molecular mechanisms of ppZP3-induced POI.

As an essential component of the ovarian follicular zona pellucida and a target antigen, ZP3 effectively induces zona pellucida-related immune injury. This model is particularly suitable for studying T cell-mediated autoimmune oophoritis in thymus-intact animals. Its strengths include high target specificity, controllability, simple methodology, and ovarian histomorphological similarity to human autoimmune POI. However, the lengthy modeling period and variability in individual immune responses may compromise stability.

### Crude ovarian antigen immunization

The ovary, a female-specific reproductive organ, contains autoantigens such as α-actinin-4, heat shock protein 70 (HSP70), and β-actin. These antigens localize to various ovarian components—including oocyte cytoplasm, follicular membranes, granulosa cells, corpus luteum, and zona pellucida—and are expressed during folliculogenesis (*Mande et al., 2011*). As ovarian antigens are tissue-specific (*Ownby & Shivers, 1972*), they can trigger autoimmune responses by inducing anti-ovarian antibodies (*Damjanović & Janković, 1989*; *Sharif et al., 2019*; *Tuohy & Altuntas, 2007*), thereby mimicking ovarian dysfunction caused by autoantibodies or T-cell-mediated immune attacks in patients.

As early as 1989, researchers homogenized rat ovarian tissue in phosphate buffer saline (PBS), emulsified it with Freund's adjuvant (1:1 ratio) to a final antigen concentration of 250 mg/mL, and administered 100 μL *via* subcutaneous injection. This induced anti-ovarian antibody production, reduced follicular counts, and suppressed fertility in rats (*Damjanović & Janković, 1989*). Similar studies using bovine or rat ovarian homogenates emulsified with Freund's adjuvant in rodents demonstrated ovarian inflammation, activated B/T cells in germinal centers, elevated serum anti-ovarian antibodies appeared after 28 days, and impaired fertility, histology showed immune cell infiltration (*Sharif et al., 2019*; *Tuohy & Altuntas, 2007*), confirming the role of crude ovarian antigens in experimental autoimmune oophoritis (EAOO).

*Wang et al. (2020)* established a rat POI model using crude ovarian antigens to evaluate the therapeutic effects of human umbilical cord mesenchymal stem cells (hUC-MSCs). The standardized protocol is summarized in Fig. 2 as below: extracting total ovarian proteins (200 mg/mL) from rats; emulsifying the proteins with Freund's adjuvant (1:1 ratio); administering 0.35 mL of the emulsion *via* subcutaneous multi-point injections every 10 days for three cycles. Incomplete Freund's adjuvant was used for the second and third immunizations (Fig. 2).

Compared to antigen-specific models like ZP3 immunization—which primarily drive B-cell-mediated autoantibody production against zona pellucida proteins—crude ovarian antigen immunization employs polyclonal antigen mixtures to induce broader autoimmune pathology, including T-cell dysregulation and multi-component inflammation, making it suitable for studying autoantibody-mediated pathology or pro-inflammatory mechanisms involving T cells and their cytokines. While most autoimmune diseases involve autoantibodies, the specific ovarian antigens responsible for human autoimmune POI remain unidentified due to the complexity and heterogeneity of crude ovarian antigen

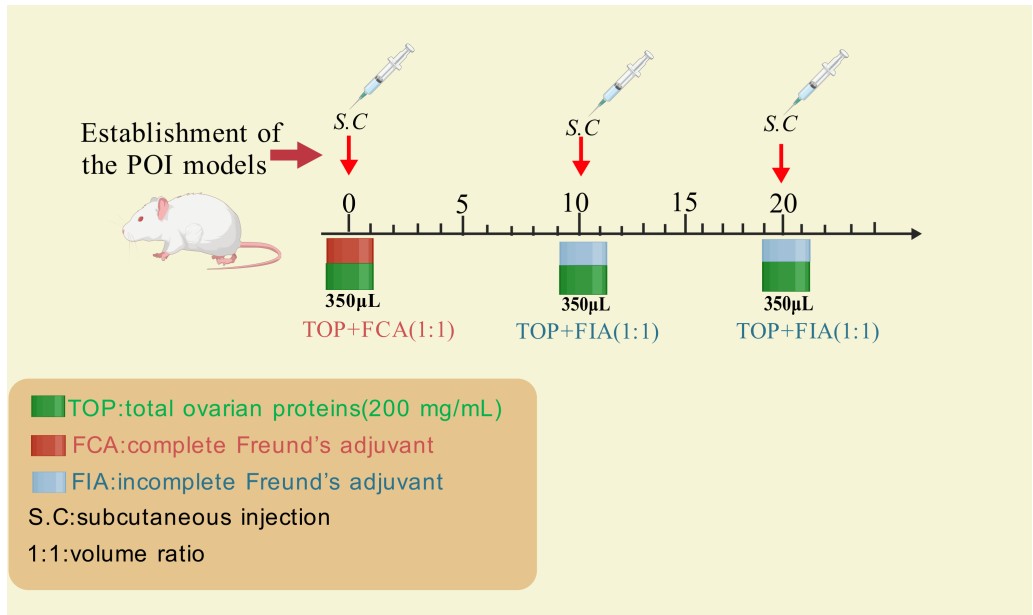

**Figure 2** **The method for inducing POI using ovarian crude antigens.** Created with BioRender.com.

preparations (*Pouladvand et al., 2024*; *Tong & Nelson, 1999*). Additionally, the effective concentration of ovarian antigens required to successfully establish autoimmune POI models remains unclear, further limiting the widespread application of this method.

### pZP4

The recombinant porcine zona pellucida 4 (pZP4) antigen is synthesized and purified, then dissolved in PBS to prepare a pZP4 solution. Similar to the ZP3 immunization protocol, the pZP4 solution is emulsified with an equal volume of Freund's complete or incomplete adjuvant. Mice are immunized *via* subcutaneous multi-point injection of 0.1 mL emulsion containing five µg pZP4. Post-immunization, the mice exhibit immune-mediated POI features, including prolonged estrous cycles, reduced serum estradiol levels, elevated anti-pZP4 antibody titers, and decreased ovarian follicles and corpora lutea (*Tang et al., 2013*). While both pZP4 and ZP3 models target zona pellucida proteins to induce autoimmune ovarian damage, key differences exist in their immunological profiles and research applications, pZP4 offers superior antigen specificity but lacks the comprehensive T-cell activation observed in whole-tissue antigen models (*Tang et al., 2013*). Therefore, this model is less frequently utilized compared to ZP3-based approaches, potentially due to limited research on pZP4-specific immune mechanisms.

## Thymectomy

The thymus, situated in the anterior mediastinum near the sternum, is a critical central immune organ in humans. Its primary cell types include lymphocytes, epithelial cells, and dendritic cells, which collectively support immune functions by serving as the site for T-cell development, differentiation, and maturation, as well as a hub for self-immune tolerance

(*Sauce & Appay, 2011*). Additionally, the thymus secretes various hormones involved in immune regulation and plays significant roles in anti-aging, anti-infection, and anti-tumor activities (*Boehm & Swann, 2013*; *Dinges et al., 2024*). Following thymectomy, thymic output, total T-cell count, T-cell subsets, and humoral immune function decline, leading to impaired immune surveillance and homeostasis, which disrupts immune balance and may trigger systemic immune responses (*Zlamy et al., 2016*).

Historically, thymic function was overlooked, as animal models (typically using adult subjects) showed no immune defects after thymectomy, likely because the thymus in mature animals does not generate plasma cells or germinal centers during normal immune responses (*Miller, 2020*). However, studies on neonatal thymectomy (NTx) revealed severe consequences. Compared to sham-operated (STX) controls, NTx mice exhibited lymphocyte deficiency, increased susceptibility to viral infections and tumors, and loss of antigen-specific immune responses (*Miller, 2020*; *Miller, 1961*; *Miller, Grant & Roe, 1963*; *Miller, De Burgh & Grant, 1965*). During murine development, lymphocytes first emerge in the thymus and later populate peripheral tissues (*e.g.*, spleen, lymph nodes, gut-associated lymphoid tissue), with thymic cortical lymphocyte proliferation surpassing that of other lymphoid organs (*Miller, 2020*; *Miller, 1961*; *Miller, Grant & Roe, 1963*; *Miller, De Burgh & Grant, 1965*; *Miller, 1966*). Thus, thymectomy at or shortly after birth causes systemic T-cell depletion, immune tolerance imbalance, and compromised immune responses, predisposing to infections (*Miller & Osoba, 1967*).

Neonatal thymectomy in mice or rats leads to systemic T-lymphocyte depletion, resulting in multi-organ autoimmune diseases such as autoimmune thyroiditis, autoimmune oophoritis, and ulcerative colitis (*Kosiewicz & Michael, 1990*). Within 1–5 months post-surgery, anti-oocyte/anti-zona pellucida antibodies are detectable in serum (*Tung et al., 1987*), followed by complete loss of oocytes and follicles in adulthood (*Taguchi et al., 1980*). Approximately 90% of thymectomized animals develop autoimmune oophoritis and ovarian failure (*Kojima & Prehn, 1981*; *Taguchi et al., 1980*). Strain-specific susceptibility varies: 100% of SWRAF1, 90% of A/J, and 35% of BALB/cBy neonatal mice exhibit autoimmune ovarian inflammation and functional decline after thymectomy (*Tung et al., 1987*; *Sakaguchi, Takahashi & Nishizuka, 1982*). In (C57BL/6 Crx A/J)F1 mice, thymectomy initially manifests as irregular estrous cycles and localized mononuclear cell infiltration around growing follicles. By adolescence, this progresses to extensive mononuclear infiltration, ovarian atrophy, complete destruction of primordial and growing follicles, and circulating autoantibodies (*Miyake et al., 1988*).

Thymectomy for constructing POI animal models must be performed within 2–4 days postpartum (typically day 3). This model induces systemic immune deficiencies, leading to organ-specific autoimmune attacks on the ovaries, thyroid, and other tissues accompanied by autoantibody production and multi-glandular autoimmune disorders (*Nelson, 2001*). The model-induced multi-glandular autoimmune disease shows strong similarity to human autoimmune POI, and the histological distribution of ovarian lymphocyte infiltration is also similar to that in humans, both characterized by reduced natural killer cell activity (*Maity et al., 1997*) and defective regulatory T cells (*Tung et al., 1987*).

Therefore, this model is particularly suitable for simulating the pathogenesis of female autoimmune POI and developing specific diagnostic and therapeutic methods. However, since thymectomy should be completed within 2–4 days after birth, the surgical technique requirements for establishing the model through thymectomy are high, the surgery is difficult, and the mortality rate of mice is high, making it still challenging to widely promote and apply at present.

## Inhibin-$\alpha$-targeted disruption of the pituitary-ovarian axis

Inhibin, predominantly synthesized by ovarian granulosa cells (*Mayo, 1994*), acts as a critical regulator of the pituitary-ovarian axis by suppressing follicle-stimulating hormone (FSH) release and counterbalancing activin to modulate ovarian function (*Hillier & Miró, 1993*; *Halvorson & De Cherney, 1996*). Both inhibin and activin belong to the transforming growth factor-βsuperfamily. While activin is composed of homo- or heterodimers of β-subunits (*Bloise et al., 2019*), inhibin A and B are heterodimeric glycoproteins formed by an α-subunit covalently linked *via* disulfide bonds to either βA or ββ subunits (*Vale et al., 1988*). Experimental evidence indicates that activin A administration during the proestrus phase in rats elevates serum FSH levels, whereas inhibin A suppresses FSH secretion during proestrus and estrus while augmenting estradiol concentrations in metestrus and diestrus—a mechanism implicating inhibin in follicular maturation regulation (*Woodruff et al., 1993*).

Functionally, inhibin exerts selective negative feedback on pituitary FSH synthesis, promotes follicular development, and serves as a paracrine mediator of FSH production (*Gregory & Kaiser, 2004*). By antagonizing activin receptor signaling, inhibin suppresses activin-driven FSH synthesis, oocyte maturation, and ovulation (*Gray, Bilezikjian & Vale, 2002*; *Li et al., 2018*). This dual action reduces FSH-mediated follicular recruitment and growth, thereby preserving the primordial follicle pool, maintaining follicular homeostasis, and delaying POI. Autoimmune targeting of inhibin-α disrupts this equilibrium by inducing neutralizing antibodies that impair FSH regulation, resulting in pathological FSH elevation, accelerated follicular depletion.

Inhibin A is secreted primarily by dominant follicles and luteal cells, whereas inhibin B originates from small primary and secondary follicles (*Hillier & Miró, 1993*). Genetic ablation of inhibin-α in mice leads to near-complete penetrance of hypergonadotropic FSH elevation (*Matzuk et al., 1992*). A clinical correlation between inhibin and POI was identified in a patient with a chromosomal translocation (46,XX,t[2;15][q32.3;q13.3]), where the breakpoint on chromosome 2 disrupted the inhibin-α gene locus (2q33-qter) (*Burton et al., 2000*). Pathogenic variants in inhibin-α protein domains or promoter regions may impair FSH suppression, driving FSH hyperactivation and POI (*Shelling et al., 2000*; *Harris et al., 2005*). In premenopausal women, INHA expression inversely correlates with basal FSH levels and age, serving as a biomarker of declining ovarian reserve (*Danforth et al., 1998*). Furthermore, INHA subunit mutations increase activin bioavailability while reducing functional inhibin, collectively elevating FSH and predisposing to POI (*Marozzi et al., 2002*).

Autoimmune Targeting Mechanism: immunization of SWXJ mice with inhibin-α-derived p215-234 peptide activates CD4+ Th1 cells exhibiting a polarized cytokine profile (high IFN-γ andIL-2, low IL-5 and IL-10), triggering B-cell production of inhibin-α-neutralizing antibodies. Immunohistochemical analyses revealed progressive CD3+ T-cell infiltration in perifollicular regions at 8 and 12 weeks post-immunization, confirming cell-mediated autoimmune involvement. Th1-derived cytokines (IFN-γ, IL-2) recruit macrophages and cytotoxic T cells, which directly damage granulosa cells. This dual attack—both cellular (T-cell infiltration) and humoral (neutralizing antibodies) blocks activin's suppressive effect on FSH, leading to sustained FSH elevation, prolonged metestrus/diestrus phases, superovulation, and accelerated primordial follicle depletion—culminating in Experimental Autoimmune Oophoritis (EAO) (*Altuntas, Johnson & Tuohy, 2006*). This model exhibits a biphasic phenotype: transient hyperfertility due to excessive follicular recruitment, followed by irreversible ovarian failure resembling human POI. Mice with high-titer inhibin-α-neutralizing antibodies recapitulate hallmark features of human disease, including FSH dysregulation, diminished ovarian reserve, and progressive fertility loss. The model now explicitly links Th1-driven autoimmunity to ovarian failure, bridging hormonal and immune dysregulation in POI pathogenesis. The trajectory—a transient FSH-driven compensatory phase (distinct from human POI) followed by ovarian collapse—partially highlights the model's utility for dissecting FSH-driven autoimmune mechanisms and testing therapeutic interventions targeting this pathway, despite temporal differences from human POI progression. The mechanism of inhibin-α action is illustrated in Fig. 3.

Although this model effectively mimics hormonal feedback dysregulation, its translational utility is constrained by phenotypic disparities in early disease stages, exceptionally prolonged administration (8–12 weeks, substantially extending standard POI modeling protocols), and the multifactorial etiology of human POI, which involves genetic, autoimmune, and environmental factors beyond inhibin-α signaling alone.

## Gene-edited models
### RAG knockout models

The adaptive immune system achieves antigen recognition through recombination-activating genes (RAG1 and RAG2), which encode endonucleases that initiate the combinatorial joining of variable (V), diversity (D), and joining (J) gene segments in T-cell receptors (TCRs) and immunoglobulins (*Notarangelo et al., 2016*; *Kenter, Priyadarshi & Drake, 2023*). This process generates diverse T- and B-cell repertoires capable of recognizing a broad spectrum of antigens. During V(D)J recombination, RAG1 and RAG2 bind to recombination signal sequences (RSSs)—conserved heptamer and nonamer motifs flanking V, D, and J segments—separated by 12- or 23-nucleotide spacers (*Fugmann et al., 2000*). These lymphocyte-specific proteins form a tetrameric complex in developing T and B cells, introducing double-strand DNA breaks at RSS-coding junctions (*Grawunder & Harfst, 2001*; *Feeney, Goebel & Espinoza, 2004*). Subsequent repair *via* non-homologous end joining completes the assembly of functional antigen receptor genes (*Notarangelo et al., 2016*).

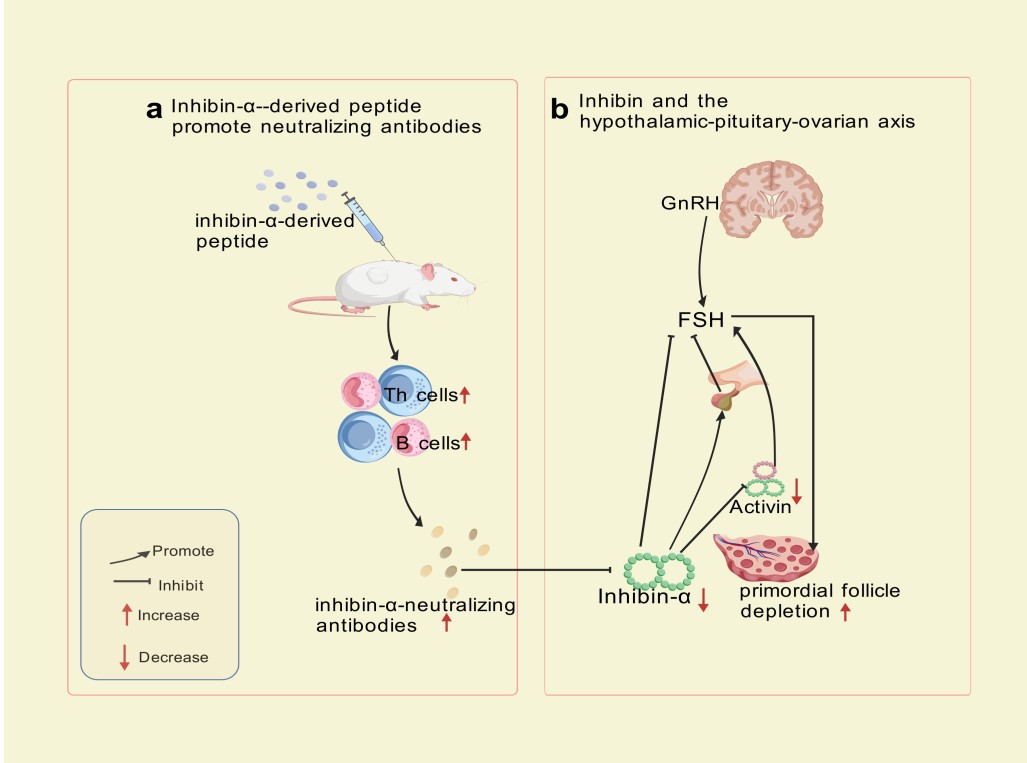

**Figure 3** **Inhibin-α-targeted disruption of the pituitary-ovarian axis.** Created with BioRender.com.

Human RAG deficiencies are associated with a spectrum of immune dysregulations, including severe combined immunodeficiency (SCID), Omenn syndrome, autoimmune cytopenias, and vasculitis (*Farmer et al., 2019*; *Geier et al., 2020*). Null mutations in RAG1 or RAG2 abolish V(D)J recombination, leading to arrested T- and B-cell development, while hypomorphic mutations permit residual activity, manifesting as atypical immunodeficiencies, granulomatous disease, or delayed-onset autoimmunity (*Delmonte, Schuetz & Notarangelo, 2018*; *Niehues, Perez-Becker & Schuetz, 2010*; *Villa et al., 1998*; *Villa et al., 2001*).

In Rag1$^{-/-}$ or Rag2$^{-/-}$ mice, the absence of V(D)J recombination blocks T-cell maturation at the CD4−CD8− double-negative stage and halts B-cell development at the pro-B (B220+CD43+) phase, resulting in thymic atrophy, splenic hypoplasia, and complete loss of mature lymphocytes (*Delmonte, Schuetz & Notarangelo, 2018*; *Mombaerts et al., 1992*; *Shinkai et al., 1992*; *Spanopoulou et al., 1994*; *Diamond et al., 1997*). Notably, these mice retain natural killer (NK) cell populations and exhibit normal viability, making them ideal for testing immune reconstitution therapies (*e.g.*, Treg transfer) and targeted immunomodulators in POI (*Bosticardo, Pala & Notarangelo, 2021*).

To investigate Treg deficiency-driven TH1 responses in POI, Xue et al. adoptively transferred CD4+CD25−45RBhi T cells ($4\times10^5$ cells/mouse) into Rag1$^{-/-}$ mice. Within 5 weeks, recipients developed POI features: reduced ovarian size, diminished follicular
reserves, decreased estradiol/progesterone levels, elevated pro-inflammatory cytokines, and increased granulosa cell apoptosis. This model uniquely permits the dissection of TH1-specific effects without confounding adaptive immune responses, with outcomes replicated in Rag2$^{-/-}$ mice, confirming that Treg deficiency triggers TH1-mediated ovarian dysfunction *via* GC apoptosis and steroidogenic impairment (*Jiao et al., 2021*).

Rag$^{-/-}$ models, characterized by absent adaptive immunity and disrupted immune homeostasis, are widely used in xenotransplantation, oncology, vaccine development, autoimmune disorders, and graft-versus-host disease research (*Villa & Notarangelo, 2019*). Critically, Rag$^{-/-}$ models serve as invaluable tools for informing therapeutic strategies in POI, particularly for patients with regulatory T-cell (Treg) deficiencies. Their utility extends to: (1) testing Treg therapies for POI patients, (2) studying immune reconstitution's impact on ovarian reserve, and (3) screening TH1-targeting biologics. Though limited by cost/focus on immunodeficiency, emerging gene-editing tools expand their potential for developing POI treatments, particularly for Treg-deficient cases. However, their application in autoimmune POI remains limited, primarily due to the high costs of gene-editing technologies and a predominant focus on immunodeficiency mechanisms.

## AIRE knockout models

The autoimmune regulator (AIRE) gene, a master transcriptional regulator of central immune tolerance, was identified in 1997 as the causative gene for autoimmune polyendocrine syndrome type 1 (APS-1) through genetic screening (*Nagamine et al., 1997*). Located on human chromosome 21q22.3 (mouse chromosome 10), AIRE encodes a 545-amino acid protein containing four conserved domains: caspase recruitment domain (CARD), Sp100, AIRE, NucP41/75 and Deaf1 (SAND) domain, and two plant homeodomains (PHD1/2) (*Nagamine et al., 1997*; *Blechschmidt et al., 1999*; *Finnish-German APECED Consortium, 1997*). The CARD facilitates homomultimerization, a process disrupted by missense mutations clustered in APS-1 patients (*Halonen et al., 2004*).

AIRE governs thymic epithelial differentiation by directing medullary thymic epithelial cells (mTECs) to express tissue-restricted antigens (TRAs). This promotes negative selection of autoreactive T cells and fosters regulatory T cell (Treg) development (*Yang et al., 2015*), thereby maintaining immune tolerance and preventing autoimmunity.

AIRE is also expressed in ovarian tissue. Genetic screening of 48 Hungarian POI patients (aged 15–39) using next-generation sequencing revealed AIRE as a susceptibility locus (*Illés et al., 2024*). Aire-knockout (KO) mice recapitulate human multi-organ autoimmunity, featuring lymphocytic infiltration, circulating autoantibodies, and infertility (*Ramsey et al., 2002*). Ovarian pathology progresses with age: lymphocyte infiltration emerges by 4 weeks, dramatic follicular depletion occurs by 8 weeks, and >50% of mice exhibit complete follicular loss by 20 weeks, with residual ovarian tissue showing eosinophilic deposits and lymphocyte absence (*Anderson et al., 2002*; *Warren et al., 2014*).

AIRE deficiency disrupts thymic TRA expression, unleashing autoreactive T cells that attack ovarian tissue. Aire-KO mice exhibit enhanced peripheral T-cell proliferation, reduced Tregs, accelerated follicular atresia, and ovarian inflammation—mirroring APS-1

features such as oophoritis, follicular reserve exhaustion, and autoantibody production. While this model faithfully replicates immune-mediated ovarian decline, its utility is limited by high construction costs and confounding multi-organ pathologies.

## Other gene-edited models

Immune homeostasis relies on a complex network of regulatory genes. Targeted disruptions of Forkhead box protein P3 (Foxp3) or placental brain-derived neurotrophic factor (BDNF) may also induce autoimmune POI. Foxp3, a master transcription factor for Treg cells, is essential for maintaining immune tolerance. Reduced Treg numbers correlate strongly with POI pathogenesis (*Yin et al., 2018*; *Wang et al., 2018*). Foxp3 ablation in mice eliminates functional Tregs, predisposing to systemic autoimmunity and ovarian dysfunction. Heterozygous BDNF knockout mice survive to adulthood but develop POI-like phenotypes by 1 month of age, characterized by ovarian hypofunction and follicular depletion. This phenotype stems primarily from utero deficits: severely diminished primordial germ cell proliferation at E11.5 due to reduced placental BDNF, establishing a depleted ovarian reserve, coupled with evidence of mitochondrial dysfunction and accelerated ovarian aging postnatally. This suggests BDNF's role in maintaining ovarian reserve and steroidogenic capacity (*Liu et al., 2024*).

These models underscore the interplay between immune dysregulation and ovarian failure. However, their translational relevance is constrained by pleiotropic effects and incomplete recapitulation of human POI heterogeneity.

## Adoptive transfer nude mouse model

Nude mice, first identified in 1966, are congenital athymic mutants harboring a recessive "nu" allele. Homozygous (nu/nu) mice exhibit hairlessness, growth retardation, reduced fertility, and premature death within 5 months (*Flanagan, 1966*; *Pantelouris, 1968*). To sustain breeding, heterozygous (nu/+) females are typically crossed with homozygous males (*Kramer & Gershwin, 1976*).

These mice lack functional T cells due to thymic agenesis, resulting in severe immunodeficiency and an inability to reject xenografts or mount self-antigen-directed immune responses (*Wortis, 1971*; *Pelleitier & Montplaisir, 1975*).

Transferring normal T cells into nude mice serves as a method to induce autoimmune oophoritis, leveraging their T cell-deficient background to study immune-mediated ovarian injury. For example: transfer of neonatal splenocytes, thymocytes, or mature thymocytes from BALB/c mice into syngeneic nu/nu mice reconstitutes partial immunity and triggers autoimmune oophoritis and gastritis in >50% of recipients, accompanied by serum autoantibodies (*Smith et al., 1992*; *Taguchi et al., 1986*). Transplantation of 15-day embryonic rat thymic rudiments into BALB/c nu/nu mice generates hybrid thymic structures (donor epithelium + host lymphocytes). By 3 months post-transplant, 92.8% of recipients develop autoimmune oophoritis with complete follicular loss, mononuclear cell infiltration, and autoantibodies against oocyte cytoplasm, zona pellucida, and steroidogenic cells (*Smith et al., 1992*). Renal capsule implantation of embryonic rat thymus or neonatal BALB/c thymus similarly induces severe autoimmune oophoritis, characterized by CD4+

effector T-cell infiltration and circulating autoantibodies (*Sakaguchi & Sakaguchi, 1990*). Even adoptive transfer of anti-lyt-1-complement-treated splenocytes into nude mice suffices to provoke organ-specific ovarian autoimmunity (*Sakaguchi et al., 1985*). These findings demonstrate that pathogenic autoreactive T cells, derived from donor thymus or spleen, can breach immune tolerance in athymic hosts and mediate systemic or organ-specific autoimmunity.

This model mirrors human thymic dysplasia-associated ovarian failure, where girls with thymic hypoplasia exhibit ovarian atrophy and follicular depletion (*Miller & Chatten, 1967*). By enabling mechanistic studies of thymic dysfunction and autoimmunity, the nude mouse adoptive transfer system provides insights for diagnosing and treating immune-mediated POI. This model also circumvents technical challenges and high infection risks associated with surgical thymectomy models recapitulates adrenal autoimmunity/Addison's disease comorbidity—characterized by adrenal cortical lymphocytic infiltration (predominantly CD8+ T cells) and parenchymal atrophy—observed in POI patients (*Hellesen, Bratland & Husebye, 2018*), and avoids confounding age-related ovarian senescence seen in non-autoimmune models. However, the concurrent multi-organ autoimmune damage may obscure ovarian-specific mechanisms. High mortality and fragility of nude mice also complicate long-term studies.

## Passive transfer of autoantibodies

Passive transfer involves injecting pathogenetic autoantibodies (*e.g.*, anti-ovarian antibodies) or serum from humans/animals with immune-mediated ovarian dysfunction into recipient animals to induce ovarian injury.

In 1989, ovarian antigen-enriched supernatant from rats was used to immunize rabbits. Subsequent transfer of rabbit anti-ovarian serum to rats completely suppressed fertility and induced ovarian hypofunction, demonstrating that ovarian antigens can trigger pathogenic autoantibodies capable of inducing experimental autoimmune oophoritis (*Damjanović & Janković, 1989*).

Immune serum from Strongyloides stercoralis-infected gerbils with reduced fertility, when transferred to infected recipients during peak parasite fecundity, significantly reduced worm L1/adult ratios (*Thompson et al., 1997*). Immunization with a murine cytomegalovirus-based contraceptive vaccine expressing zona pellucida 3 (ZP3) generated ZP3-specific antibodies in mice. Passive transfer of this serum to unvaccinated BALB/c mice prolonged median time to conception (*Lloyd et al., 2010*). In myasthenia gravis studies, transfer of patient-derived immunoglobulins to healthy animals successfully recapitulated disease symptoms, validating the broader applicability of this approach (*Richman et al., 2012*).

While this model is straightforward and rapid-onset, it fails to recapitulate the multifaceted immunopathology of spontaneous autoimmunity. Variable antibody half-lives and inter-individual variability in antibody titers may compromise experimental reproducibility, limiting its widespread use.

## Candidate autoantigens in premature ovarian insufficiency

*Tong & Nelson (1999)* identified a 125-kDa cytoplasmic oocyte protein targeted by autoantibodies in D3tx-induced autoimmune POI mice. Screening an ovarian cDNA library with autoimmune serum revealed Maternal Antigen That Embryos Require (MATER), a maternal effect protein exclusively expressed in oocytes (*Tong & Nelson, 1999*; *Tong, Nelson & Dean, 2000*). MATER persists from growing oocytes to late blastocysts and is essential for post-zygotic embryonic development (*Tong et al., 2004*). MATER$^{-}$/$^{-}$ females are sterile, positioning MATER as a candidate autoantigen in human POI.

*Sundblad et al. (2006)* detected antibodies against a 50-kD ovarian antigen (identified as enolase) in 20% of POI patients, suggesting its role in polyglandular autoimmune syndromes associated with ovarian failure. 3β-Hydroxysteroid Dehydrogenase (3βHSD) is implicated in both POI (21% of patients) and type 1 diabetes (23% of patients), potentially serving as a shared autoantigen in multi-organ autoimmune disorders (*Arif et al., 1996*; *Reimand et al., 2000*). Expressed from primordial follicles to mature oocytes, HSP90β may contribute to ovarian autoimmunity by exposing immunogenic epitopes during folliculogenesis (*Pires & Khole, 2009*). Emerging reports suggest a potential link between quadrivalent HPV vaccination and POI, possibly mediated by vaccine adjuvants or cross-reactive immunity. However, clinical evidence remains limited and requires validation (*Gong et al., 2020*).

A comparative summary of the commonly used modeling methods for immune-mediated POI animal models discussed in this review is provided in Table 1. Additionally, potential ovarian antigens such as MATER, enolase, 3βHSD, and HPV4 may induce immune-mediated POI and serve as novel targets for developing POI animal models.

## Evaluation methods for successful POI modeling

Current criteria for assessing the efficacy of premature ovarian insufficiency (POI) animal models encompass a multi-dimensional evaluation framework (*Dai et al., 2023*; *Francés-Herrero et al., 2024*), including:

(i) Reproductive capacity: fertility indices (*e.g.*, total litter count, average litter size, median inter-litter intervals).

(ii) Hormonal profiles: serum levels of anti-Müllerian hormone (AMH), estradiol (E2), follicle-stimulating hormone (FSH), and luteinizing hormone (LH).

(iii) Ovarian morphology: ovarian volume, weight, and fibrosis quantification.

(iv) Follicular dynamics: primordial/antral follicle counts, regularity of estrous cycles, and post-ovulation metrics (retrieved oocyte numbers, abnormal ovulation rates).

(v) Apoptotic biomarkers: expression levels of Ki67 (proliferation marker), Bcl2 (anti-apoptotic), Bax (pro-apoptotic), Caspase 3, and Caspase 9, combined with histological evidence of granulosa cell apoptosis.

These parameters collectively reflect ovarian reserve depletion, endocrine dysfunction, and cellular turnover mechanisms, providing a robust basis for validating POI model fidelity to human pathophysiology.

**Table 1   Animal models for constructing immune-related premature ovarian insufficiency.**

| Model types | Induction methods | Targeted immune components | Advantages | Disadvantages | Applications |
|---|---|---|---|---|---|
| pZP3 | Artificial synthesis of ZP3 polypeptide as an antigen, combined with Freund's adjuvant to immunize animals, induces autoimmune responses targeting the ovarian zona pellucida, leading to oophoritis and follicular depletion. | Attack ovarian zona pellucida; T-cell-mediated autoimmune mechanisms | Straightforward procedure, high model stability, ovarian pathology closely resembles human autoimmune POI; ovary-specific targeting avoids multi-organ interference. | Requires multiple immunizations and has a lengthy modeling period. | Preferred choice for evaluating drug or stem cell therapies targeting autoimmune mechanisms, suitable for studying T-cell-mediated autoimmune mechanisms and immunomodulatory treatments. |
| Crude ovarian antigen | Utilizes whole ovarian homogenate or extracted ovarian proteins as antigens to induce anti-ovarian antibodies for targeted destruction of the ovaries. | Inducing the production of anti-ovarian antibodies | Simple procedure and low cost | Complex antigen composition with poor specificity; risk of triggering systemic immune responses, requiring caution. | Commonly applied to investigate mechanisms of polyglandular autoimmune syndrome. |
| Thymectomy | Thymectomy is performed in neonatal mice (2–4 days after birth) to disrupt T-cell development, resulting in immune tolerance disruption and multi-organ immune attacks. | The removal of the thymus results in the inability of T cells to develop | Ovarian pathology features complete follicular destruction and mononuclear cell infiltration, closely resembling human autoimmune POI. | High surgical complexity requiring microsurgical techniques ; elevated mortality rates necessitate strict protocols and specific mouse strain requirements. | Suitable for studying thymic defects and T cell-mediated autoimmune dysregulation mechanisms. |
| Inhibin-α | inhibin-α-derived p215-234 peptide induces neutralizing antibodies, interfering with its FSH regulatory role and accelerating follicle pool depletion. | Induce the production of neutralizing antibodies against inhibin α, preventing activin-induced downregulation of pituitary FSH release, thereby leading to increased FSH levels and excessive depletion of the ovarian follicle pool. | A key regulatory molecule in the pituitary-ovarian axis; its immunologically targeted disruption strongly mimics hormonal dysregulation in human POI. | Lengthy modeling period, technically challenging, costly; may trigger immune dysregulation in other endocrine glands. | Suitable for studying pituitary-specific immunotargeting therapeutic strategies and mechanisms of autoimmune antibody-related diseases. |
| AIRE Knockout | Knockout of AIRE Aire leads to enhanced proliferation of self-reactive T cells that attack ovarian tissues, alongside a reduction in Treg cells. | AIRE drives thymic epithelial cell differentiation to promote negative selection of autoreactive T cells and Treg development, maintaining central tolerance. AIRE deficiency accelerated follicular atresia, and ovarian inflammation. | The AIRE knockout model exhibits features similar to diseases like APS1 and APECED, providing a highly relevant pathological model to study ovarian decline mechanisms in these conditions. | High construction costs; ovarian damage may coexist with pathologies in other organs, complicating isolated analysis. | Suitable for investigating POI mechanisms involving multi-organ immune dysfunction. |
| Rag gene knockout combined with T-cell transfer | Knockout of Rag1 or Rag2 genes results in T/B lymphocyte deficiency and severe immune dysfunction. | Knockout of Rag genes disrupts V(D)J recombination, resulting in blocked development and complete absence of T and B cells, consequently causing severe combined immunodeficiency. | Mimics pathological features of premature ovarian insufficiency (POI). The absence of endogenous T cells eliminates host immune interference, while enabling targeted study of specific T-cell subsets. | Complex model construction requiring integration of gene-editing and cell transplantation technologies. | Ideal for dissecting immune subset imbalances and investigating pro-inflammatory signaling pathways or antibody-mediated follicular atresia in multi-organ autoimmune damage contexts. |
| Adoptive Transfer Nude Mouse Model | Adoptive transfer of T cells or T cell-producing immune organs (e.g., thymus, spleen) from normal donors into nude mice induces organ-specific autoantibodies to attack target organs, leading to multi-organ damage. | Nude mutant mice lack thymus and thymus-derived T cells. Adoptive T-cell transfer into nude mice triggers organ-specific autoantibodies attacking target tissues, resulting in multi-organ damage. | Overcomes challenges of thymectomy-based models (e.g., technical difficulty, high infection risk); serves as a robust model for POI with adrenal autoimmunity/Addison's disease. | High husbandry requirements for nude mice; technically demanding and costly T-cell isolation/purification; mouse strain variability may compromise model stability. | Suitable for exploring mechanisms and therapies for thymic hypoplasia or severe thymic defects And for investigating immune cell subset regulation. |
| Passive Transfer of Autoantibodies | Direct transplantation of anti-ovarian antibodies or immune serum containing anti-ovarian antibodies into animals leads to ovarian damage as the antibodies bind to ovarian antigens. | Anti-ovarian antibodies directly bind to ovarian antigens, inducing ovarian tissue damage and inflammation. | Simple procedure for the immune model and rapid onset of effects. | Lacks active immunization processes; fluctuations in antibody titers and individual variability may lead to inconsistent experimental results. | Rarely used in practical research. |

## CONCLUSIONS

Research on autoimmune POI is extensive, and ideal disease animal models are critical for mechanistic studies, therapeutic discovery, and drug development. These models should exhibit the following characteristics: (1) practicality: simple methodology, short timeframes, low cost, and high reproducibility. (2) Pathophysiological fidelity: high resemblance to the pathogenic mechanisms and progression observed in human POI patients. (3) Stability and specificity: consistent and targeted induction of ovarian dysfunction without confounding systemic effects. (4) Reversibility: pathological changes that can be ameliorated through therapeutic interventions. When selecting a model, the primary criteria should align with your research objectives and mechanistic focus, followed by secondary considerations such as model stability, specificity, time requirements, cost, and the technical feasibility of the modeling protocol. Ultimately, we hope this review can guide researchers in choosing appropriate and reliable POI animal models, thereby advancing future investigations into POI and its therapeutic strategies.

## ACKNOWLEDGEMENTS

Thank for our laboratory colleagues.

### Funding
This review was funded by "Science and Technology Fund Project of Guizhou Health Commission, grant number gzwkj2024-439/" and "Science and Technology Plan Project of Chinese Qianxinan Prefecture, grant number 2024-93/". The funders had no role in study design, data collection and analysis, decision to publish, or preparation of the manuscript.

### Grant Disclosures
The following grant information was disclosed by the authors:
Science and Technology Fund Project of Guizhou Health Commission: gzwkj2024-439/.
Science and Technology Plan Project of Chinese Qianxinan Prefecture: 2024-93/.

### Competing Interests
The authors declare there are no competing interests.

### Author Contributions
- Anchun Hu conceived and designed the experiments, performed the experiments, analyzed the data, prepared figures and/or tables, authored or reviewed drafts of the article, and approved the final draft.
- Yanli Mu analyzed the data, prepared figures and/or tables, and approved the final draft.
- Guanyou Huang conceived and designed the experiments, authored or reviewed drafts of the article, and approved the final draft.
- Shuyun Zhao conceived and designed the experiments, analyzed the data, authored or reviewed drafts of the article, and approved the final draft.

## Patent Disclosures

The following patent dependencies were disclosed by the authors:

The illustrations in Figs. 1–3 were generated using BioGDP.com (a publicly accessible tool for scientific visualization). All content is labeled with the attribution: "Created with BioRender.com" (Note: corrected platform name based on common practice; replace if different). We confirm that these figures do not contain copyrighted material requiring third-party permissions.

## Data Availability

This is a literature review.

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
