# Peer review of "Comparison of animal models for immune premature ovarian insufficiency"

_PeerJ, doi:10.7717/peerj.20091_

## Round 0.1 · original submission · Major Revisions

·

Basic reporting

no comment

Experimental design

no comment

Validity of the findings

no comment

Additional comments

Dear Authors,
Please provide your response to the following comments/suggestions:

1. In this manuscript, the authors mention the role of ovarian biopsy as a diagnostic gold standard, but there is no mention of current non-invasive diagnostic criteria (e.g., serum FSH levels, AMH, ultrasound) or limitations thereof, which would provide a more holistic diagnostic perspective. Please include these details briefly.
2. The final introduction introduces the article’s focus on autoimmune POI animal models but does not specify the type of model or methods (e.g., chemical induction, immunization, genetic models). Please write a brief description in the introduction, also from the reader’s point of view.
3. Please define what the controversies are in POI diagnosis or classifications.
4. Are rodent models truly representative of human autoimmune POI? Please justify.
5. What are the translational challenges from animal models to clinical therapies? Need to separate the paragraph on this discussion.
6. Survey methodology: It would be beneficial to mention and provide details regarding about timeframe of the literature search, such as 2000–2025, whether the authors follow any PRISMA or any systematic review guideline.
7. Please provide a summary table of the selected models included in the review.
8. Though authors mention several models without sufficient explanation of how they induce ovarian autoimmunity or what aspect of immune dysfunction they model. Need to provide justification.
9. Authors cited multiple references to validate the ZP3 model, but there is minimal comparison to other antigen-induced or autoimmune models. A brief description with models using crude ovarian antigens or pZP4 would strengthen the evaluative depth.
10. Section: Line 147-175- This section mentions the involvement of Th1/Th17 responses. But there is a very limited mechanistic approach mentioned. How the immune response progresses from antigen recognition to ovarian damage is not well described. Please mention these details. The mentioned immunization protocol should be clear and understandable way. A stepwise or tabular summary of antigen dose, route, frequency, and adjuvants would improve clarity.
11. Authors cited several references from the 1980s and early 2000s. There is little indication of whether or how this model has been modernized or validated against current immunological standards or technologies.
12. Section Line: 234-277: While the role of Th1 and B cells is mentioned, the section does not explore cytokine involvement (e.g., IFN-γ, IL-2) or immune cell infiltration in ovarian tissue. The autoimmune component remains underexplored compared to the hormonal aspects. Please revise.
13. Considering the model's complex hormonal and immune interactions, a figure or diagram could greatly aid understanding of it.
14. The authors mention that RAG KO models are used broadly, but do not explore how they might inform therapeutic development or immune reconstitution strategies in POI, especially in patients with regulatory T-cell deficiencies.

There are serious flaws in this manuscript. Authors need to revise this manuscript to gain a better readership.

Thank you.

Reviewer 2 ·

Basic reporting

1. Please add a figure legend to Figure 1.
2. Please carefully review and correct Table 1 to ensure accurate presentation of the information.

Experimental design

No comment.

Validity of the findings

No comment.

Additional comments

The authors perform a comprehensive review of various animal models used to study immune-mediated POI, a topic important to ovarian immunologists, reproductive endocrinologist and infertility specialists. My comments are as follows:
1. As mentioned previously, the authors should add a figure legend to Figure 1 to clearly explain the symbols, abbreviations, and any other elements within the figure.
2. The table has formatting issues that need to be addressed. For example, 1) the “DTX3” (a candidate autoantigen of POI) and “Thymectomy” are placed on the same row. 2) A statement describing the adoptive transfer nude mouse model from the main text (lines 381-385) - “circumvents technical challenges and high infection risks associated with surgical thymectomy models, recapitulates adrenal autoimmunity/Addison’s disease comorbidity observed in POI patients” - is incorrectly placed with the Rag knock out mice in Table 1.
3. Line 50-51, the sentence “ovarian biopsy remains the gold standard for POI diagnosis” has some doubts, and it might be more appropriate to say it’s the gold standard for autoimmune-POI.
4. Line 272-274, the authors claim that the early enhancement of fertility followed by a collapse in mice with inhibin-α-neutralizing antibodies is similar to clinical characteristics of human POI. However, based on my understanding, increased fertility is not typically observed in the early stages of POI in human patients. Please provide supporting evidence for this claim.
5. Line 275-278, the authors mention that the translational utility of the inhibin-α model is limited by the multifactorial etiology of human POI. Could the authors elaborate further on why they believe this model's translational application is limited?
6. Line 347-350, please provide an explanation of the immune mechanisms involved in the POI phenotype observed in BDNF heterozygous knockout mice.
7. Line 383-384, the text mentions adrenal autoimmunity/Addison's disease in relation to the adoptive transfer nude mouse model. Please provide a detailed description of the adrenal autoimmunity phenotype exhibited by this model.

---

## Round 0.2 · Minor Revisions

This revised manuscript has been much improved. However, minor revisions are still needed as suggested by reviewers.

·

Basic reporting

-

Experimental design

-

Validity of the findings

-

Additional comments

Dear Authors,
Thank you for revising the manuscript based on the comments/suggestions.
All the best.

Reviewer 2 ·

Basic reporting

-

Experimental design

-

Validity of the findings

-

Additional comments

The content of this version is much more comprehensive and complete. The readability of the manuscript has significantly improved with the enriched information in the table and the addition of new figures. The following are a few comments for the authors:

1. Please provide relevant citations to support the claims made in lines 42-45, as the diagnosis of POI based on FSH and menstrual dysfunction is generally accepted.

2. The assertion that "the ovarian antigens are tissue-specific" needs more clarification. The preceding text mentions α-actinin-4, heat shock protein 70 (HSP70), and β-actin as examples of autoantigens found in the ovary. However, these particular genes are generally considered to be ubiquitously expressed in various tissues, not exclusively in the ovary.

3. The manuscript briefly introduces pZP4 immunization as a modeling method. However, a more detailed explanation of the ZP4 gene itself is warranted. Can ZP4 be expressed in mice? Furthermore, a more in-depth discussion on why pZP4 offers "superior antigen specificity" would be beneficial.

4. Lines 323-324, stating that the trajectory is "distinct from human POI," might be overly rigid. It is important to acknowledge that some forms of POI caused by genetic mutations (e.g., PTEN) can indeed exhibit a transient period of accelerated follicular activation before premature depletion.

5. The Rag knockout models are discussed as leading to severe immunodeficiency and requiring adoptive transfer of T cells to induce ovarian phenotypes. If isolated RAG1 or RAG2 knockout alone does not inherently result in POI, and the ovarian phenotype only appears after exogenous Treg cell transfer, then it might be worth reconsidering its primary classification within "Gene-Edited Models" as a direct POI model.

---

## Round 0.3 · accepted · Accept

This revised manuscript is well organized and addresses all the concerns raised by reviewers. I am happy with the current version, which is ready for publication.